# Dissecting Gradient Masking and Denoising in Diffusion Models for Adversarial Purification

## Abstract

Diffusion models exhibit remarkable empirical robustness in adversarial purification. The mechanisms underlying such improvements remain unclear. It is possible that diffusion models effectively purify the adversarial examples via the learned stimuli prior. Alternatively, the substantial randomness added in the diffusion models may cause gradient masking that contaminates the empirical estimate of adversarial robustness. Here, we seek to dissect the contribution of these two potential factors. Theoretically, we illustrate how a purification system with randomness can cause gradient masking, which can not be addressed by the standard expectation-over-time (EOT) method. Inspired by this, we propose and justify that a simple procedure, randomness replay, can provide a better robustness estimate when randomness is involved. Experimentally, we verify that gradient masking indeed happens under previous evaluations of diffusion models. After properly controlling the effect of randomness, the reverse-only diffusion model (RevPure) provides a similar robustness improvement with the previous DiffPure framework, suggesting that the robustness improvement is solely attributed to the reverse process. Furthermore, our analyses reveal that robustness improvement is caused by a sequential denoising mechanism that transforms the stimulus to a direction orthogonal to the original adversarial perturbation, rather than reducing the $\ell_2$ distance between the transformed and clean stimuli. Our results shed new light on the mechanisms underlying the empirical robustness from diffusion models, and shall inform future development of more efficient adversarial purification systems.

## 1 Introduction

Neural networks are vulnerable to small adversarial perturbations (Szegedy et al., 2013; Goodfellow et al., 2014), which presents a fundamental question on the robustness of artificial learning systems. Adversarial training (Madry et al., 2017), has become the most successful method to overcome this problem (Shafahi et al., 2019; Pang et al., 2020; Wang et al., 2021). However, research has found that training with a specific attack usually sacrifices the robustness against other types of perturbations (Schott et al., 2018; Ford et al., 2019; Yin et al., 2019), indicating that adversarial training simply overfits the attack rather than achieving an overall robustness improvement.

Adversarial purification provides an alternative path toward adversarial robustness. This approach typically relies on generative models to purify the stimulus before passing to a classifer (Song et al., 2017; Samangouei et al., 2018; Shi et al., 2021; Yoon et al., 2021). The basic idea is to leverage the stimuli prior learned by generative models to project adversarial perturbations back towards the stimuli manifold. Intuitively, the performance of such purification should depend on how well the generative models capture the probability distribution of natural stimuli. Recently, adversarial purification based on diffusion models(Ho et al., 2020; Song et al., 2020b) (DiffPure) was reported to show promising improvements against various attacks on multiple datasets (Nie et al., 2022). Diffusion models consist of a forward diffusion process and a reverse process performing sequential denoising. Notably, both steps inject substantial randomness. It is well known that randomness can induce gradient masking (Papernot et al., 2017; Athalye et al., 2018) when evaluating the adversarial robustness with gradient-based attacks (Carlini et al., 2019). This creates an inherent challenge for evaluating the adversarial robustness of diffusion models.

Thus, previously reported empirical robustness from diffusion models Nie et al. (2022) may consist of both (i) *bona fide* improvement due to its ability in denoising and (ii) gradient masking. Through

a combination of theoretical and empirical analyses, we dissect the contribution of the two in the empirical robustness of diffusion models. While we focused on adversarial purification based on diffusion models, our results may have broad implications in other types of adversarial defense that involve randomness or denoising. Our main contributions are summarized below:

- Theoretically, we show that a purification system with randomness can cause gradient masking that cannot be solved by expectation-over-time (EOT) (Athalye et al., 2018), thus providing a false sense of robustness in previous protocols (Sec. 4.2). We propose "randomness replay", which can better estimate the robustness for systems involving randomness (Sec. 4.3).

- Our empirical results corroborate our theoretical predictions, thus, randomness-induced gradient masking happens in DiffPure (Sec. 5.1). Experiemnts also show that the forward process is not critical for robustness—reverse-only diffusion models (RevPure) lead to a similar robustness improment (Sec. 5.2).

- We identify the mechanisms underlying the adversarial robustness in diffusion models. Specifically, the reverse process sequentially denoises an adversarial example to an orthogonal direction relative to the adversarial direction, rather than reducing the $L_2$ distances to the clean stimulus (Sec. 6).

## 2 RELATED WORK

**Diffusion models for adversarial purification**    Diffusion models (Ho et al., 2020; Song et al., 2020b) set the SOTA performances on image generation, and represent a natural choice for adversarial purification. Nie et al. (2022) proposed the DiffPure framework, which utilized both the forward and reverse process and achieved promising empirical robustness comparable with adversarial training on multiple benchmarks. Similar improvements were reported with guided diffusion models (Wang et al., 2022). These studies led to substantial interest in applying diffusion models for adversarial purification in various domains, including auditory data (Wu et al., 2022) and 3D point clouds (Sun et al., 2023). However, none of the works systematically studied and evaluated the randomness and the potential gradient masking effect within diffusion models. The mechanism of empirical robustness improvement was also not well understood.

Another line of research applies diffusion models to improve certified robustness Cohen et al. (2019). Carlini et al. (2022) found that plugging diffusion models as a denoiser into the denoised smoothing framework (Salman et al., 2020) can lead to non-trivial certified robustness. Xiao et al. (2023) further developed this method and explained the improvement in certified robustness.

**Gradient masking and randomness**    Gradient masking has been defined as "construct a model that does not have useful gradients" (Papernot et al., 2017). It may provide a false sense of robustness against gradient-based attacks (Tramèr et al., 2018). Athalye et al. (2018) further identified that randomness could cause gradient masking as "stochastic gradients", and proposed the expectation-over-time (EOT) which became the standard evaluation for stochastic gradients. EOT is often required to rule out the effect of gradient masking for a fair evaluation of robustness (Carlini et al., 2019).

## 3 PRELIMINARIES

**Adversarial purification**    Adversarial purification intends to first "purify" the perturbed data (or image) before classification. Consider a purification system $P(\boldsymbol{x})$, which processes perturbed data $\tilde{\boldsymbol{x}}$ close to the clean data $\boldsymbol{x}$, and further been readout by a classifier $F(\boldsymbol{x}) = \boldsymbol{y}$. Under the assumption that adversarial purification satisfies $P(\tilde{\boldsymbol{x}}) \approx \boldsymbol{x}$, the Bypass Direct Approximation (BPDA) (Athalye et al., 2018) can provide a robustness estimation if $P$ is hard-to-differentiate.

**Randomness in diffusion models**    Diffusion models gradually disassemble an arbitrary distribution into a standard Gaussian, therefore inherently contain randomness in the forward and reverse process. The forward process of Denoising Diffusion Probabilistic Models (DDPM) (Ho et al., 2020) is

$$\boldsymbol{x}_t = \sqrt{\alpha_t}\boldsymbol{x}_{t-1} + \sqrt{1 - \alpha_t}\boldsymbol{\epsilon}, \ \boldsymbol{\epsilon} \sim \mathcal{N}(\boldsymbol{0}, \boldsymbol{I}), \tag{1}$$

in which the $\epsilon$ will introduce randomness. Further, the reverse process

$$\boldsymbol{x}_{t-1} = \frac{1}{\sqrt{\alpha_t}} \left( \boldsymbol{x}_t - \frac{1 - \alpha_t}{\sqrt{1 - \bar{\alpha}_t}} \boldsymbol{\epsilon}_\theta(\boldsymbol{x}_t, t) \right) + \sigma_t \boldsymbol{z}, \ \boldsymbol{z} \sim \mathcal{N}(\boldsymbol{0}, \boldsymbol{I}) \tag{2}$$

also introduces randomness through $\boldsymbol{z}$. Notably, deterministic reverse process has also been proposed, *e.g.,* in Denoising Diffusion Implicit Models (DDIM) (Song et al., 2020a) the reverse process

$$\boldsymbol{x}_{t-1} = \sqrt{\bar{\alpha}_{t-1}}\hat{\boldsymbol{x}}_0 + \sqrt{1 - \bar{\alpha}_{t-1}}\boldsymbol{\epsilon}_\theta(\boldsymbol{x}_t, t), \ \hat{\boldsymbol{x}}_0 = \frac{\boldsymbol{x}_t - \sqrt{1 - \bar{\alpha}_t}\boldsymbol{\epsilon}_\theta(\boldsymbol{x}_t, t)}{\sqrt{\bar{\alpha}_t}} \tag{3}$$

is fully deterministic and thus does not introduce randomness.[1]

## 4 RANDOMNESS-INDUCED GRADIENT MASKING

### 4.1 PROBLEM SETUP

For an $n$-dimensional data point $\boldsymbol{x}_0 \in \mathbb{R}^n$ with label $\boldsymbol{y}_0$, we study the classification performance under perturbations within an $\ell_p$ ball $\mathbb{B}(\boldsymbol{x}_0, \epsilon)$ with radius $\epsilon$. For any perturbations beyond the ball, we clip it back following the usual practice of adversarial attacks. We first make the following definitions for a classification system $S$.

**Definition 1** (Absolute robustness). *The absolute robustness of $S$ around $\boldsymbol{x}_0$ within ball $\mathbb{B}$ is given by*

$$R(S, \boldsymbol{x}_0, \mathbb{B}) = \mathbf{1}(S(\boldsymbol{x}) = \boldsymbol{y}_0, \ \forall \boldsymbol{x} \in \mathbb{B}).$$

Thus, we define it as 1 if it is a perfect robust classification system, and 0 if otherwise there exists an adversarial point.

**Definition 2** (Empirical adversarial robustness). *For some adversarial examples generate by attacking method $\boldsymbol{\xi} \sim \Xi$, the empirical adversarial robustness is given by*

$$\tilde{R}(S, \boldsymbol{x}_0, \mathbb{B}, \Xi) = \Pr\left[S(\boldsymbol{x_0} + \boldsymbol{\xi}) = \boldsymbol{y}_0\right] = 1 - \Pr\left[S(\boldsymbol{x_0} + \boldsymbol{\xi}) \neq \boldsymbol{y}_0\right].$$

**Definition 3** (Empirical adversarial attack failure). *For the non-robust case $R(S, \boldsymbol{x}_0, \mathbb{B}) = 0$, if the empirical adversarial robustness $\tilde{R}(S, \boldsymbol{x}_0, \mathbb{B}, \Xi)$ significantly overestimates the robustness, we identify it as attack failure. It can be quantified by the attack failure rate*

$$\Delta = \tilde{R}(S, \boldsymbol{x}_0, \mathbb{B}, \Xi) - R(S, \boldsymbol{x}_0, \mathbb{B}) = \begin{cases} \Pr\left[S(\boldsymbol{x_0} + \boldsymbol{\xi}) = \boldsymbol{y}_0\right], & R(S) = 0 \\ 0, & R(S) = 1 \end{cases}.$$

*Smaller failure rate $\Delta$ means better estimation of absolute robustness based on the empirical attack.*

Note that gradient masking is a special case of attack failure—when the attack is gradient-based, and the attack failure is caused by manipulating the gradient to be non-optimal.

While the above definitions are based on the point case, they could be generalized to the dataset case by taking expectations, which is similar to the definition used in Viallard et al. (2021). The definition 1 may appear to be strict at first glance. Below we explain why it is indeed an appropriate definition. Consider we run an adversarial attack on a dataset and get an attack success rate of 80%. Assume the attack is perfect, thus we can always find an adversarial example if there exists one in the region, which is the ultimate goal for adversarial attack research. Then the accuracy means for 80% of the data, we manage to find at least one adversarial example within the region. Thus, the empirical robustness with a perfect attack is a good approximation of the absolute robustness. Imperfect attacking will affect the probability of finding the example (Def. 2), but will not altering the absolute robustness as an inherent property of the data and classification system.

### 4.2 RANDOMNESS-INDUCED GRADIENT MASKING

Following the setup in Sec. 4.1, consider a classifier $F(\boldsymbol{x})$ with a simply connected adversarial space $\mathbb{A} = \{\boldsymbol{x} | F(\boldsymbol{x}) \neq \boldsymbol{y}_0, \boldsymbol{x} \in \mathbb{B}(\boldsymbol{x}_0, \epsilon)\}$. Denote the centroid of $\mathbb{A}$ as $\boldsymbol{c} = \int_\mathbb{A} \boldsymbol{x} d\boldsymbol{x} / \int_\mathbb{A} d\boldsymbol{x}$, the radius of $\mathbb{A}$ as $r = \max_{\boldsymbol{x} \in \mathbb{A}} \|\boldsymbol{x} - \boldsymbol{c}\|_2$. Consider the additive Gaussian noise model

$$P(\boldsymbol{x}) = \boldsymbol{x} + \boldsymbol{\eta} \sim \mathcal{N}(\boldsymbol{0}, \sigma^2 \boldsymbol{I}). \tag{4}$$

---

[1]We follow the notation of the DDPM paper, thus the form is slightly different from the DDIM paper. The $\bar{\alpha}_t$ in DDPM is corresponding to the $\alpha_t$ in DDIM.

For a perfect attack (thus can always find an adversarial example if exists), denote the generated adversarial perturbation as $\boldsymbol{\xi}$, expectation-over-time (EOT) of the perturbation as $\mathbb{E}\boldsymbol{\xi}$. Then the success rates of applying such perturbations to the new classification system with randomness $S = P \circ F$ are given by the following theorem.

**Theorem 1** (Success rate of perfect attack with randomness). *For $S = P \circ F$,*

$$\Pr[S(\boldsymbol{x}_0 + \boldsymbol{\xi}) \neq \boldsymbol{y}_0)] \leq \Psi\left(\frac{r^2}{2\sigma^2}; n\right), \ \Pr[S(\boldsymbol{x}_0 + \mathbb{E}\boldsymbol{\xi}) \neq \boldsymbol{y}_0)] \leq \Psi\left(\frac{r^2}{\sigma^2}; n\right),$$

*where $\Psi(x; n)$ is the cdf of chi-square distribution $\chi^2(n)$.*

**Corollary 1.1** (Curse of dimensionality). *When $n \to \infty$, $r/\sigma \ll \sqrt{n}$,*

$$\lim_{n \to \infty} \Pr[P \circ F(\boldsymbol{x}_0 + \boldsymbol{\xi}) \neq \boldsymbol{y}_0)] = \lim_{n \to \infty} \Pr[P \circ F(\boldsymbol{x}_0 + \mathbb{E}\boldsymbol{\xi}) \neq \boldsymbol{y}_0)] = 0.$$

See Appendix A for the proof. An illustration of the proof is shown in Fig. S1a. Note that the assumption of simply connected adversarial space is appropriate: Tramèr et al. (2017) showed that the adversarial space spans a continuous region, where indeed exists orthogonal bases such that their linear combinations are still adversarial examples with high probability.

The theorem shows that the failure of adversarial attacks involving randomness is essentially a problem of high-dimensional space. As illustrated in Fig. S1c, the cdf of $\chi^2(n)$ distribution inclines to be flat as the dimensionality $n$ rises. The upper bound of the success rate for the non-EOT case $\Psi(r^2/2\sigma^2)$ is strictly smaller than the success rate for the EOT case $\Psi(r^2/\sigma^2)$, which explains the slight benefits of applying EOT in robustness evaluation with randomness. However, the proof illustrates that the key problem lies in the noise during testing $\boldsymbol{\eta}_1$, but not during attacking. The EOT can only handle noise in attacking $\boldsymbol{\eta}_0$. The non-EOT and EOT methods have no significant differences as they all approach zero in the high-dimensional case.

The theorem explains why the Yoon et al. (2021) model only observed signification robustness improvement against BPDA-EOT after injecting Gaussian noise into the system, as well as Byun et al. (2020) reported robustness against black-box attacks after applying a small amount of Gaussian noise. Indeed, we can make a non-robust system impossible to attack under the previous evaluation protocol (but still non-robust) by simply adding Gaussian noise.

### 4.3 MORE ACCURATE ROBUSTNESS ESTIMATION USING RANDOMNESS REPLAY

To fairly evaluate the absolute robustness, one needs to cancel the effect of noise during testing (such as removing $\boldsymbol{\eta}_1$). If randomness is an inherent property of the system (such as diffusion models), one can achieve this by either (i) the future randomness during testing $\boldsymbol{\eta}_1$ and compensating it in attacking, which is not possible for non-pseudo noise, or (ii) during testing, replay the exact same randomness encountered in generating the attack, thus $\boldsymbol{\eta}_1 = \boldsymbol{\eta}_0$, to ensure that the calculated attack is still optimal in this particular model randomness configuration. A more formal description of randomness replay is given in Algorithm 1.

---

**Algorithm 1** Randomness Replay

**Input:** A system with randomness $S$, data $\boldsymbol{x}$, adversarial attack algorithm $\Xi$, memory buffer $\mathcal{M}$
**Output:** Empirical robustness estimation of $S$ at point $\boldsymbol{x}$

Attacking:                                              Testing:
Generate attack $\boldsymbol{\xi} \sim \Xi(S(\boldsymbol{x}; \boldsymbol{\eta}))$ with random-    Retrieve $(\boldsymbol{x}, \boldsymbol{\eta}, \boldsymbol{\xi})$ from $\mathcal{M}$
ness $\boldsymbol{\eta}$                                 **return** $S(\boldsymbol{x} + \boldsymbol{\xi}; \boldsymbol{\eta})$
Store $(\boldsymbol{x}, \boldsymbol{\eta}, \boldsymbol{\xi})$ in $\mathcal{M}$

---

One may be concerned that whether randomness replay would alter the behavior of the model, and thus, change the robustness of the system. To address this potential concern, we show that randomness replay does not change the absolute robustness of the system.

**Theorem 2** (Equivalence of randomness replay). *For any system with randomness $S$, and its randomness replay version $S'$,*

$$R(S, \boldsymbol{x}_0, \mathbb{B}) \equiv R(S', \boldsymbol{x}_0, \mathbb{B}).$$

It shows that randomness replay is just an alternative scheme for calculating/applying adversarial attacks, without changing the properties of the system such as absolute robustness. We immediately get the following corollary formally states randomness replay can make an accurate robustness estimation for the system stated in Sec. 4.2.

**Corollary 2.1** (Effectiveness of randomness replay). *For the system $S$ stated in Sec. 4.2, after applying randomness replay, the success rate of gradient-based attack is*

$$\Pr[S'(\boldsymbol{x}_0 + \boldsymbol{\xi}) \neq \boldsymbol{y}_0)] = R(S, \boldsymbol{x}_0, \mathbb{B}).$$

To sum up, we gain the following insights from the theoretical analysis:

- Gradient masking can happen in a system with randomness, which is caused by the testing noise $\boldsymbol{\eta}_1$ rather than the attacking noise $\boldsymbol{\eta}_0$.
- EOT would produce a better robustness estimation by canceling the effect of $\boldsymbol{\eta}_0$, but not fundamentally solving the problem.
- By randomness replay, we can eliminate the effect of such randomness-induced gradient masking, and produce a more accurate robustness estimation.
- EOT may not provide any further benefits after applying randomness replay.

Next, we will examine whether such predictions hold in diffusion models for adversarial purification.

## 5 EXPERIMENTS

### 5.1 RANDOM-INDUCED GRADIENT MASKING IN DIFFUSION MODELS

**Randomness replay by noise fixing**   We first apply the randomness replay to examine whether randomness-induced gradient masking happens by the DiffPure method Nie et al. (2022). Ideally, we should store the randomness encountered while calculating the attack, and replay it during testing once meet the same stimuli. In practice, we found it more convenient to just control the random seed to be the same during attacking and testing for the same stimuli. For the $i$-th batch of data at the $t$ step of the forward/reverse process, we set the random seed

$$\mathtt{Seed}(i, t) = \begin{cases} 2 \times (1000 \times i + t) + 1, & \text{if forward process} \\ 2 \times (1000 \times i + t), & \text{if reverse process} \end{cases} \tag{5}$$

before sampling the Gaussian noise from eq. 1 or eq. 2. This setting ensures that we have a different random seed for each batch of data and timesteps in the forward/reverse process, but will keep the randomness the same through the entire purification process if encountering the same data batch.

**Gradient masking in DiffPure**   We evaluate the robustness improvements against BPDA and BPDA-EOT attacks (Athalye et al., 2018) on CIFAR-10 with $\ell_\infty = 8/255$, and $\ell_\infty = 4/255$ on ImageNet, which are standard evaluations for adversarial purification methods. We used the official check-point of DDPM on CIFAR-10 as the purification system, and a standard WideResNet-28-10 on CIFAR-10 as the classifier. We used the entire dataset for BPDA, and 10% of the dataset for BPDA-EOT. We use 15 EOT samples—the same as Nie et al. (2022). For ImageNet experiments, we used the official checkpoint of $256 \times 256$ unconditional Guided diffusion (Dhariwal & Nichol, 2021) as the purification system, and ResNet-50 for classification. We evaluate the clean accuracy with 1000 samples from the validation set, and robust accuracy with 200 samples due to computational efficacy.

Table 1: Randomness-induced gradient masking in diffusion models.

| Purification Model | Method | Fix Noise | $t^*$ | Clean | BPDA | BPDA-EOT |
|---|---|---|---|---|---|---|
| DDPM (CIFAR-10) | DiffPure | ✗ | 100 | 84.51 | 82.38 | 80.8 |
| | | ✓ | 100 | 85.25 | 60.62 | 60.7 |
| Guided (ImageNet) | DiffPure | ✗ | 150 | 69.0 | 67.0 | – |
| | | ✓ | 150 | 68.5 | 34.0 | – |

As shown in Table 1, on CIFAR-10, our implementation of DDPM[2] with DiffPure achieves 80.8% against BPDA-EOT, which is comparable with the original result 81.40% (Nie et al., 2022). After applying randomness replay, there is no significant change in the clean accuracy, but the robust accuracy (BPDA and BPDA-EOT) will receive a 20% drop. On ImageNet, we achieve a similar performance on the clean accuracy (69.0% vs. originally 67.79%). Again, randomness replay will not affect the clean accuracy, but provide a 33% lower robust estimation against BPDA. The 34% BPDA robustness is even lower than the originally reported full gradient AutoAttack method (40.93%). Note that without randomness replay, EOT provides a slightly lower (better) robustness estimation than non-EOT, but the difference is subtle and not comparable with randomness replay. Those phenomena are precisely consistent with our theoretical predictions. All combined together, we conclude that randomness-induced gradient masking indeed happened on both CIFAR-10 and ImageNet datasets in the previous evaluation of diffusion models for adversarial purification.

**Randomness largely determines the purified states**    We further examine how strong can randomness alters the purified state for diffusion models. We considerd two scenarios: with the same initial state and different randomness, and different initial states within a small $\ell_2$ ball but with exactly the same randomenss. The correlation matrices of the eventual purified directions are shown in Fig. 1. With different noises, the diffusion model purifies the same state into diffrent (but positively correlated) directions. However, with the same noise, the diffusion model purifies different initial states within a small ball into almost the same directions (correlation > 0.99). This shows that if the perturbation of initial states is small, randomness is a decisive factor for the evantual purified state. If not properly controlling the randomness, the gradients calculated for the previous randomness will be applied to an alternative purified state and non-optimal. This is an extra piece of evidence for randomness-induced gradient masking can happen in diffusion models.

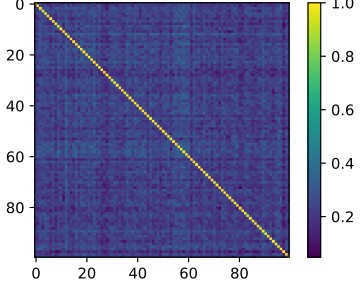
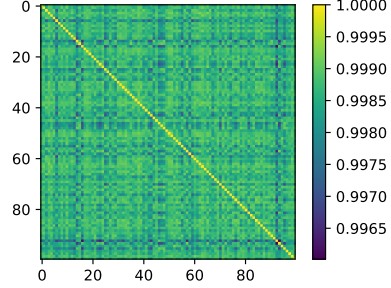

(a) Same initial state, different noise.       (b) Different initial states, same noise.

Figure 1: Randomness largely determines the purified states of diffusion models. Correlation matrices of the purified state directions with (a) same initial state, different noises. With different noises, the diffusion model purifies the same state into diffrent (but positively correlated) directions. (b) Different initial states, same noise. Provided with the same noise, the diffusion model purifies different initial states within a small ball into almost the same directions (correlation > 0.99). This shows that if the perturbation of initial states is small, randomness is a decisive factor for the evantual purified state.

## 5.2   COMPARISON OF DIFFPURE AND REVERSE-ONLY DIFFUSION MODELS (REVPURE)

**Reverse-only diffusion models (RevPure)**    The DiffPure (Nie et al., 2022) framework proposed to utilize both the forward and reverse processes of diffusion models for adversarial purification. Since the forward process introduces a large amount of randomness, we want to explore whether it's possible to remove the forward process, thus only using the reverse process of diffusion models for adversarial purification (RevPure). A similar reverse-only framework was proposed in DensePure (Xiao et al., 2023), but further equipped with a majority voting mechanism to study the certificated robustness.

**DDIM**    Besides randomness replay, an alternative way to eliminate the effect of randomness is to use a deterministic reverse process. As shown in eq. 3, the reverse process of DDIM Song et al.

---

[2]We focus on discrete-time diffusion models in this paper because there may be extra sources of gradient masking beyond randomness in continuous-time models due to numerical solvers (Huang et al., 2022).

(2020a) does not involve any randomness. Thus, by applying the reverse-only DDIM, we can get a fully deterministic diffusion model-based adversarial purification system.

Table 2: The comparison of DiffPure and reverse-only diffusion models (RevPure).

| Purification Model | Method | Fix Noise | $t^*$ | Clean | BPDA | BPDA-EOT |
|---|---|---|---|---|---|---|
| DDPM (CIFAR-10) | DiffPure | ✓ | 150 | 79.61 | 68.00 | 67.2 |
| | RevPure | ✓ | 150 | 81.92 | 74.20 | 74.9 |
| DDIM (CIFAR-10) | RevPure | – | 150 | 75.81 | 44.12 | 44.7 |
| Guided (ImageNet) | DiffPure | ✓ | 150 | 68.5 | 34.0 | – |
| | RevPure | ✓ | 100 | 57.1 | 32.0 | – |

As shown in Table 2, the RevPure framework can provide a better clean and robustness accuracy than DiffPure on CIFAR-10. The deterministic method DDIM does not exhibit comparable robustness improvements. For ImageNet, there is a decrease in clean accuracy by RevPure (see Fig. 2c). This can be explained as the reverse process of the diffusion model on ImageNet fails to perform normally on clean stimuli, thus will require the forward diffusion process to inject appropriate noise. However, the robust performances of DiffPure and RevPure are still similar. In all, the reverse denoise process in diffusion models seems to play a more fundamental role in adversarial purification.

### 5.3 THE EFFECT OF SEQUENCE LENGTH AND SUBSEQUENCING

**Sequence length**   We modified the sequence length ($t^* = 1, 50, 100, 150, 200$) and measured the robustness of DDPM with DiffPure, RevPure (with randomness replay, same for further evaluations) and DDIM against BPDA on 10% of CIFAR-10. The results are shown in Fig. 2a. The clean accuracy decreases monotonically with the sequence length. The robustness accuracy first increases with sequence length, until reaching the optimal, and further decreases as the decreasing of clean accuracy becomes the dominant effect. For both DDPM and DDIM with RevPure, the optimal robustness sequence length is at around $t^* = 150$, while for DDPM with DiffPure, the optimal is attained at $t^* = 200$ and will further decrease (not shown). Fig. 2a shows that DDPM with RevPure outperforms DiffPure for both clean accuracy and robustness. The DDIM model performs worst. On ImageNet, the effects are similar except that the clean accurary of RevPure decreases considerably faster than DiffPure, as shown in Fig. 2c.

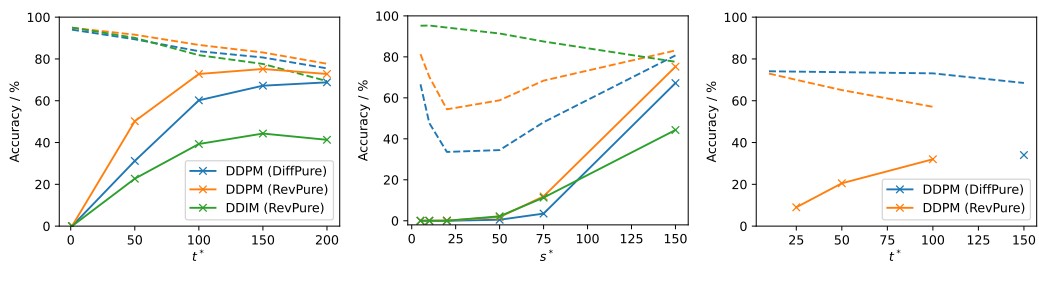

(a) Sequence length (CIFAR-10).   (b) Subsequencing (CIFAR-10).   (c) Sequence length (ImageNet).

Figure 2: The effect of sequence length and subsequencing. (a) Sequence length $t^*$. Clean accuracy decreases monotonically with the sequence length. Robustness increases with the sequence length until reaching the optimal, and then decreases as the drop of clean accuracy dominates. (b) Subsequence length $s^*$. Subsequencing deteriorates robustness. (c) Sequence length on ImageNet. The effects are similar on ImageNet, except that the clean accurary of RevPure decreases considerably faster than DiffPure. Dashed lines mark the clean accuracy and solid lines mark the BPDA accuracy.

**Subsequencing**   We further introduced the subsequencing technique for acceleration to explore its effects on robustness. For a purification model with a sequence length $t^*$, we picked a subsequence with length $s^* = 5, 10, 20, 50, 75$. We used a linear scheduler for the subsequencing. As shown in Fig.2b, the clean accuracy of DDPM first decreases and then increases with the decrease of the subsequence length. This can be explained as there are two effects influencing the accuracy:

subsequencing itself as an approximation deteriorates accuracy, while a shorter sequence length benefits accuracy. For the DDIM model, the clean accuracy increases with subseqeuncing, reflecting the fact that DDIM is a better model for subsequence than DDPM. However, as for the robustness performance, subseqeuncing significantly decreases the BPDA accuracy, showing that the robustness improvement relies on adequate iterations of the dynamics.

### 5.4 FID SCORE MAY NOT SIGNIFICANTLY AFFECT ROBUSTNESS IMPROVEMENT

We next investigate whether the image generation ability of diffusion models determines their robustness improvements for adversarial purification. Since the exponential moving average (EMA) is a critical trick of diffusion models for high-quality image generation, we simply tested the robustness of DDPM models with and without EMA during training. As shown in Fig. 3, the EMA model performs better than the non-EMA model for both clean classification and BPDA attack, which is aligned with their image generation ability. However, given their relatively large difference in image generation ability (FID score 3.212 vs. 12.138), their robustness difference (BPDA 71.57% vs. 67.47%) is less significant. This is consistent with the intuition that a low-quality but clean image may already be sufficient for the classifier to perform image classification. Thus, improvements in the FID score of diffusion models may not lead to commensurate improvement in robustness.

Table 3: Comparison of robustness for models with different image generation abilities.

| Purification Model | Method | Fix Noise | $t^*$ | FID | Clean | BPDA |
|---|---|---|---|---|---|---|
| DDPM (EMA) | RevPure | ✓ | 150 | 3.212 | 81.92 | 74.20 |
| DDPM (Non-EMA) | RevPure | ✓ | 150 | 12.138 | 80.15 | 70.21 |

## 6 HOW DO THE DIFFUSION MODELS IMPROVE ROBUSTNESS?

To gain a deeper understanding of the mechanisms of how diffusion models improve robustness, we investigate how the purified states evolve over time. Denote $x_0$ as the clean image, $\tilde{x}_0$ as the perturbed image, and $x_t$ as the purified image at timestep $t$. We quantify the change relative purified state vector $x_t - x_0$ with the initial BPDA perturbation.

**The reverse process is not a simple $\ell_2$ denoiser** We first measure the length of the vector ($\|(x_t - x_0)\|_2$). As shown in Fig. 3a, the DiffPure framework exhibits a clear two-stage process—first increase the $\ell_2$ distance by forward process and further decrease by reverse. RevPure also exhibits an increase-decrease trend. However, the $\ell_2$ distances can not explain the robustness improvement—compared to the initial adversarial perturbation, the end distances increase for all models, and the worst performing model DDIM has the least distance.

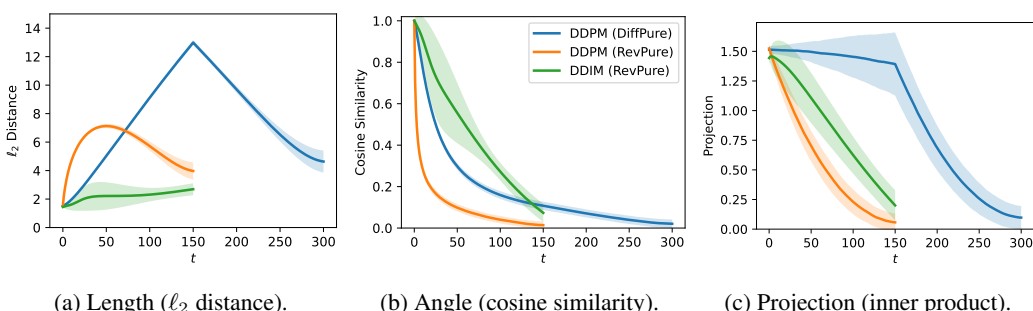

(a) Length ($\ell_2$ distance).     (b) Angle (cosine similarity).     (c) Projection (inner product).

Figure 3: The evolution of relative purified state vector $x_t - x_0$ over time on CIFAR-10. (a) Length. $\ell_2$ distance between the purified states $x_t$ and clean stimuli $x_0$. (b) Angle. Cosine similarity between the adversarial direction $\tilde{x}_0 - x_0$ and the purified direction $x_t - x_0$. (c) Projection. The inner product of $\tilde{x}_0 - x_0$ and $x_t - x_0$. The projections onto the adversarial direction, but not the $\ell_2$ distances, reflect robustness improvements from different models.

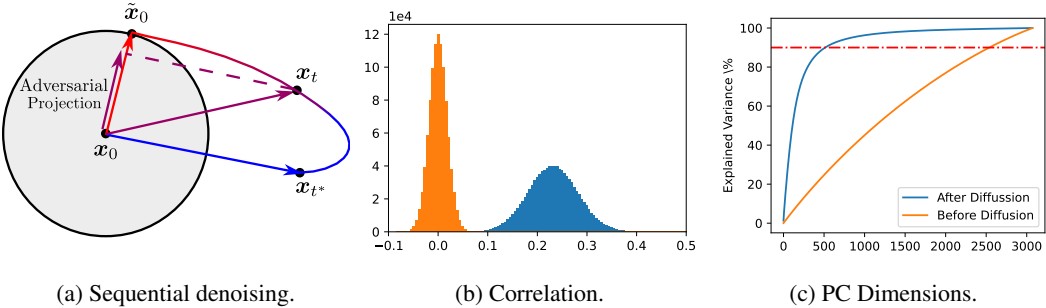

(a) Sequential denoising.      (b) Correlation.      (c) PC Dimensions.

Figure 4: Sequential denoising removes the adversarial projection. (a) An illustration of the sequential denoising in reverse process. (b) The hisotogram of correlations of initial perturbations and purified directions. Initially, the perturbation are non-correlated. After the diffusion model, the purified states are positively correlated around a direction. This illustrates that the diffusion model has an internal bias direction from image priors. (c) The PCA analysis of initial perturbations (before diffusion) and purified directions (after diffusion). It shows that the purified directions lie in a much lower dimensional space comparing with before purification. The dashed dot line marks 90% of accumulated exlained variance.

**Sequential denoising removes the adversarial projection**     We further measure the cosine similarity and inner product between the relative adversarial vector $\tilde{x}_0 - x_0$ and the relative purified state vector $x_t - x_0$, as shown in Fig.3b and 3c. An illustration of the sequential denoising process is shown in Fig. 4a. The reverse process indeed gradually removes the projection on the original adversarial direction, eventually reaching the direction that is near-orthogonal to the adversarial direction. These results further support the idea that forward process may not be useful for robustness. It also explains why we find shorter sequence lengths do not achieve significant robustness improvements as the fewer steps do not remove the adversarial projections sufficiently.

**Diffusion models purify the states toward a biased direction in a low-dimensional space**     We finally analyse the structure of purified directions. We generated 30,000 random initial perturbations aorund a clean CIFAR-10 stimulus (which is a order greater than the data dimension), and recorded the corresponding purified directions after passing the diffusion model. As shown in Fig. 4b, the initial perturbations before diffusion models are non-correlated. After the diffusion model, the purified states are positively correlated around a direction. Further PCA analysis (Fig. 4c) illustrates that the purified directions lie in a much lower dimensional space. In all, this illustrates that the diffusion model has an internal bias direction from image priors. This biased direction is unlikely to be adversarial directions, as otherwise if the diffusion models purifiy towards adversarial directions, we should witness a major decrease in clean accurary even without any attack. The low dimensional structure encodes the image prior learned form the dataset, which will filter out abnormal perturbations and thus, make the system harder to attack.

## 7   Conclusion

In this paper, we carefully studied the empirical robustness improvement from adversarial purification with diffusion models. The key is to properly understand the effect of randomness on the evaluation of empirical adversarial robustness. Leveraging a simple theoretical example, we illustrate how a purification system with randomness can cause gradient masking that cannot be solved by the standard EOT method. The theoretical arguments explain why some of the previous work relies on injecting random noise to observe "robustness improvements (Yoon et al., 2021), and question whether the promising empirical robustness from diffusion models is gradient masking (Nie et al., 2022). We propose randomness replay to provide a better robustness estimation of any randomness-involved systems. Experiments confirm that gradient masking happens in diffusion models with the previous protocol. Reverse-only diffusion models (RevPure) indeed provide a similar robustness improvements with the DiffPure framework. Further analyses show that the reverse process improves robustness by sequentially removing adversarial projections. The purified directions are postively correlated, and lie in a lower dimensional space, refelcting the image prior learned by diffusion models. A more efficient adversarial projection removal mechanism, rather than the generation ability (FID score), should lead to a better adversarial purification system.

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

# A PROOFS

## A.1 PROOF OF THEOREM 1

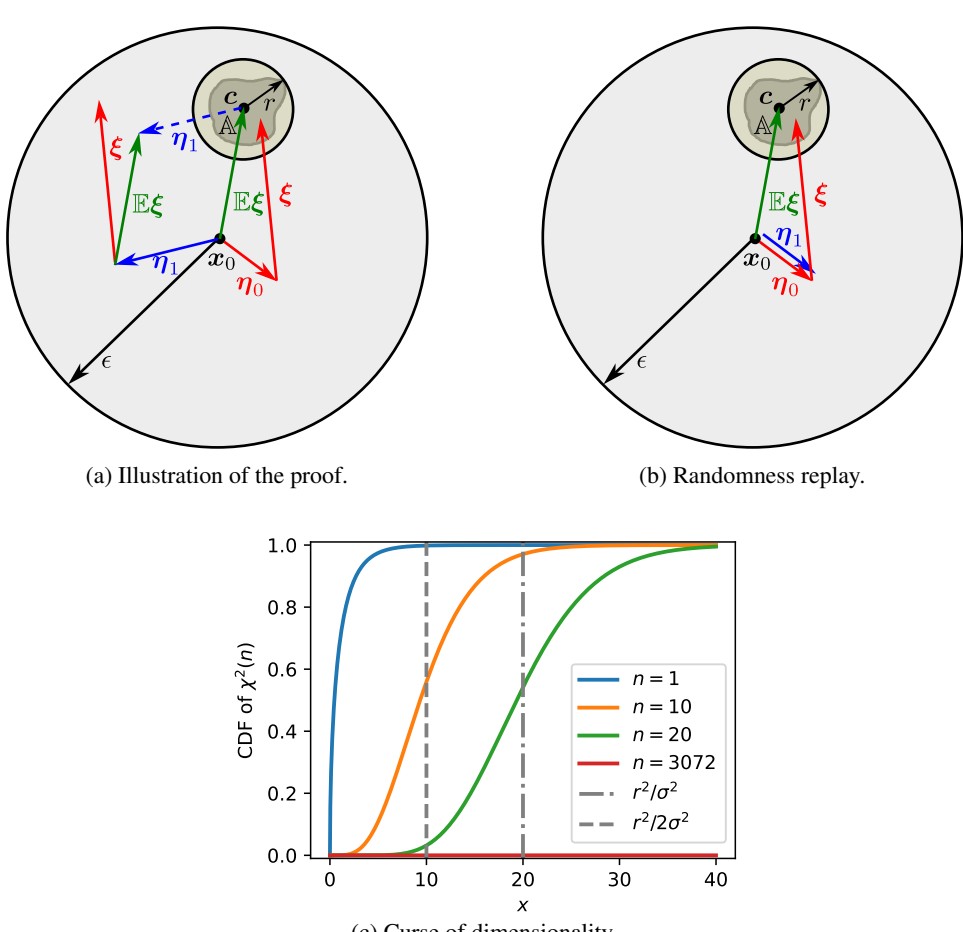

(a) Illustration of the proof.  (b) Randomness replay.

(c) Curse of dimensionality.

Figure S1: Random-induced gradient masking. (a) Illustration of the example and proof. The region considered is a $\ell_2$ hypoerball around $\boldsymbol{x}_0$ with radius $\epsilon$. The simply connected adversarial region $\mathbb{A}$ can be covered by a hyperball around $\boldsymbol{c}$ with radius $r$. EOT of the gradient-based attack $\mathbb{E}\boldsymbol{\xi}$ is able to cancel the effect of noise during generating the attack $\boldsymbol{\eta}_0$, thus pointing to the optimal direction $\boldsymbol{c} - \boldsymbol{x}_0$. The success rate of applying such attack with testing noise $\boldsymbol{\eta}_1$ is effectively the possibility of $\boldsymbol{\eta}_1$ does not escape the small $r$ ball. (b) Randomness replay. After randomness replay, since $\boldsymbol{\eta}_1 = \boldsymbol{\eta}_0$, the attack is optimal again, thus pointing back to the adversarial region. (c) Curse of dimensionality. The cdfs of $\chi^2(n)$ distribution becomes flat as the dimensionality $n \to \infty$. This would yield the gradient-based and EOT attacks approaching probability zero to success when $r/\sigma \ll \sqrt{n}$. The critical point is at $(n, 1/2)$.

*Proof.* Since $\boldsymbol{c}$ is the centroid of the adversarial space $\mathbb{A}$ with a radius of $r = \max_{\boldsymbol{x} \in \mathbb{A}} \|\boldsymbol{x} - \boldsymbol{c}\|_2$, the adversarial space can be covered by a hyperball, $\mathbb{A} \subseteq \mathbb{B}_2(\boldsymbol{c}, r)$.

Assume $P$ generates noise $\boldsymbol{\eta}_0$ while calculating the attack, then the generated attack is

$$\boldsymbol{\xi} = \boldsymbol{x}_1 - (\boldsymbol{x}_0 + \boldsymbol{\eta}_0), \forall \boldsymbol{x}_1 \in \mathbb{A}. \tag{6}$$

The EOT of the attack

$$\mathbb{E}\boldsymbol{\xi} = \boldsymbol{c} - \boldsymbol{x}_0, \tag{7}$$

which points to the centroid of the adversarial space. During testing, assume $P$ generates noise $\boldsymbol{\eta}_1$. The probability of the EOT attack successfully fools the system

$$\Pr[P \circ F(\boldsymbol{x}_0 + \mathbb{E}\boldsymbol{\xi}) \neq \boldsymbol{y}_0)] = \Pr[F(\boldsymbol{c} + \boldsymbol{\eta}_1) \neq \boldsymbol{y}_0] = \Pr[(\boldsymbol{c} + \boldsymbol{\eta}_1) \in \mathbb{A}] \leq \Pr[(\boldsymbol{c} + \boldsymbol{\eta}_1) \in \mathbb{B}_2(\boldsymbol{x}_1, r)], \tag{8}$$

which is upper bounded by the probability that after perturbation the attack is still within the hyperball. Thus,

$$\Pr[\|\boldsymbol{\eta}_1\|_2 \le r] = \Pr\left[\sum_{i=1}^n \eta_{1i}^2 \le r^2\right] = \Pr\left[\sum_{i=1}^n \left(\frac{\eta_{1i}}{\sigma}\right)^2 \le \frac{r^2}{\sigma^2}\right]. \tag{9}$$

Since the normalized $i$-th component $\eta_{1i}/\sigma \sim \mathcal{N}(0,1)$, the LHS is the sum of square of standard Gaussians, $\sum_{i=1}^n (\eta_{1i}/\sigma)^2 \sim \chi^2(n)$, therefore the probability is the cdf of $\chi^2(n)$, thus

$$\Pr[P \circ F(\boldsymbol{x}_0 + \mathbb{E}\boldsymbol{\xi}) \ne \boldsymbol{y}_0)] \le \Psi\left(\frac{r^2}{\sigma^2}; n\right). \tag{10}$$

For the non-EOT case,

$$\Pr[P \circ F(\boldsymbol{x}_0 + \boldsymbol{\xi}) \ne \boldsymbol{y}_0)] = \Pr[\boldsymbol{x}_1 + (\boldsymbol{\eta}_1 - \boldsymbol{\eta}_0) \in \mathbb{A}] \le \Pr[\boldsymbol{x}_1 + (\boldsymbol{\eta}_1 - \boldsymbol{\eta}_0) \in \mathbb{B}_2(\boldsymbol{c}, r)]. \tag{11}$$

Since $(\boldsymbol{\eta}_1 - \boldsymbol{\eta}_0) \sim \mathcal{N}(\boldsymbol{0}, 2\sigma^2 \boldsymbol{I})$, the probability

$$\Pr[P \circ F(\boldsymbol{x}_0 + \boldsymbol{\xi}) \ne \boldsymbol{y}_0)] \le \Pr[\|(\boldsymbol{x}_1 - \boldsymbol{c}) + (\boldsymbol{\eta}_1 - \boldsymbol{\eta}_0)\|_2 \le r]$$
$$\le \Pr[\|(\boldsymbol{\eta}_1 - \boldsymbol{\eta}_0)\|_2 \le r] = \Psi\left(\frac{r^2}{2\sigma^2}; n\right). \tag{12}$$

$\square$

## A.2 PROOF OF COROLLARY 1.1

*Proof.* By CLT, when $n \to \infty$, the cdf of $\chi^2(n)$ becomes the cdf of $\mathcal{N}(n, 2n)$, thus

$$\Pr[P \circ F(\boldsymbol{x}_0 + \mathbb{E}\boldsymbol{\xi}) \ne \boldsymbol{y}_0)] \le \Phi\left(\frac{r^2}{\sigma^2}; n, 2n\right) = \frac{1}{2}\left(1 + \mathrm{erf}\frac{\frac{r^2}{\sigma^2} - n}{2\sqrt{n}}\right) \approx \frac{1}{2}\left(1 - \mathrm{erf}\frac{\sqrt{n}}{2}\right), \tag{13}$$

where $\Phi(x; \mu, \sigma)$ denotes the cdf of Gaussian distribution. By the squeeze theorem, we have

$$\lim_{n \to \infty} \Pr[P \circ F(\boldsymbol{x}_0 + \mathbb{E}\boldsymbol{\xi}) \ne \boldsymbol{y}_0)] = 0, \tag{14}$$

and similarly,

$$\lim_{n \to \infty} \Pr[P \circ F(\boldsymbol{x}_0 + \boldsymbol{\xi}) \ne \boldsymbol{y}_0)] = 0. \tag{15}$$

$\square$

## A.3 PROOF OF THEOREM 2

*Proof.* If $R(S) = 1$, by definition, $\forall \boldsymbol{x} \in \mathbb{B}$, $S(\boldsymbol{x}) = \boldsymbol{y_0}$. Since $S$ and $S'$ have the same domain, and will produce the same outputs with the same inputs and randomness, $\forall \boldsymbol{x} \in \mathbb{B}$, $S'(\boldsymbol{x}) = \boldsymbol{y_0}$. Thus, $R(S') = 1$.

If $R(S) = 0$, then $\exists \tilde{\boldsymbol{x}}_0$ and a randomness configuration $\boldsymbol{\eta}$, s.t. $S(\tilde{\boldsymbol{x}}_0; \boldsymbol{\eta}) \ne \boldsymbol{y_0}$. Again, since $S$ and $S'$ have the same domain, and will produce the same outputs with the same inputs and randomness, if encounter the same randomness and inputs, $S'(\tilde{\boldsymbol{x}}_0; \boldsymbol{\eta}) = S(\tilde{\boldsymbol{x}}_0; \boldsymbol{\eta}) \ne \boldsymbol{y_0}$. Thus, $R(S') = 0$. $\square$

## A.4 PROOF OF COROLLARY 2.1

As shown in Fig. S1b, after randomness replay, since $\boldsymbol{\eta}_1 = \boldsymbol{\eta}_0$, the attack is optimal again, thus pointing back to the adversarial region. Therefore $\Pr[S'(\boldsymbol{x}_0 + \boldsymbol{\xi}) \ne \boldsymbol{y}_0)] = R(S', \boldsymbol{x}_0, \mathbb{B}) = R(S, \boldsymbol{x}_0, \mathbb{B})$.

# B    ADDITIONAL DATA

Table S1: The effect of sequence length $t^*$ (raw data for figure 2a).

| Metric | Model | $t^*$ | 1 | 50 | 100 | 150 | 200 |
|--------|-------|-------|-----|-----|-----|-----|-----|
| **Clean** | DDPM (DiffPure-Replay) | | 94.0 | 89.4 | 83.7 | 80.7 | 75.5 |
| | DDPM (RevPure-Replay) | | 94.9 | 91.6 | 86.7 | 83.1 | 77.7 |
| | DDIM (RevPure) | | 95.0 | 90.1 | 81.8 | 77.6 | 69.4 |
| **BPDA** | DDPM (DiffPure-Replay) | | 0.0 | 31.2 | 60.2 | 67.2 | 68.8 |
| | DDPM (RevPure-Replay) | | 0.0 | 50.2 | 72.8 | 75.2 | 72.8 |
| | DDIM (RevPure) | | 0.0 | 22.7 | 39.3 | 44.3 | 41.3 |

Table S2: The effect of subsequence length $s^*$ (raw data for figure 2b).

| Metric | Model | $s^*$ | 5 | 10 | 20 | 50 | 75 | 150 |
|--------|-------|-------|-----|-----|-----|-----|-----|-----|
| **Clean** | DDPM (DiffPure-Replay) | | 66.6 | 47.7 | 33.6 | 34.5 | 48.1 | 80.7 |
| | DDPM (RevPure-Replay) | | 81.3 | 70.3 | 54.4 | 58.8 | 68.4 | 83.1 |
| | DDIM (RevPure) | | 95.2 | 95.3 | 94.3 | 91.4 | 87.5 | 77.6 |
| **BPDA** | DDPM (DiffPure-Replay) | | 0.0 | 0.0 | 0.0 | 0.5 | 3.5 | 67.2 |
| | DDPM (RevPure-Replay) | | 0.0 | 0.0 | 0.1 | 1.8 | 11.9 | 75.2 |
| | DDIM (RevPure) | | 0.0 | 0.0 | 0.0 | 2.2 | 11.3 | 44.3 |

Table S3: Robustness correspondence of classifier and adversarial projection (raw data for figure **??**).

| Model | Adv. Projection | BPDA | PGD-$\ell_2$ | $\epsilon$ | Model |
|-------|-----------------|------|--------------|------------|-------|
| DDPM (DiffPure-Replay) | $0.0972 \pm 0.0985$ | 68.00 | 61.75 | 0.0972 | |
| DDPM (RevPure-Replay) | $0.0571 \pm 0.0758$ | 74.20 | 78.50 | 0.0571 | WideResNet-28-10 |
| DDIM (RevPure) | $0.2001 \pm 0.1280$ | 44.12 | 26.49 | 0.2001 | |

# C VISUALIZATION OF THE PURIFICATION PROCESS

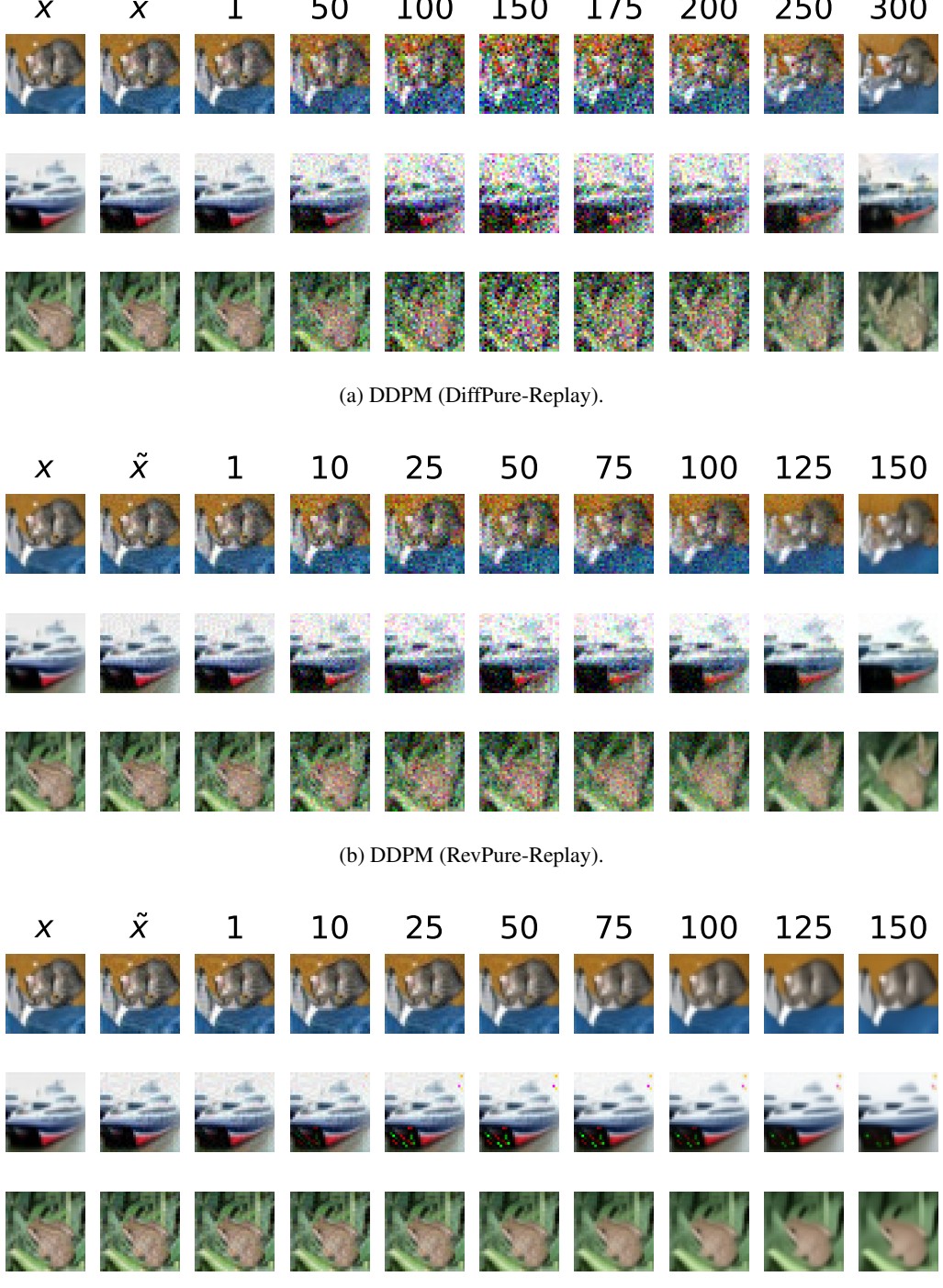

(a) DDPM (DiffPure-Replay).

(b) DDPM (RevPure-Replay).

(c) DDIM (RevPure).

Figure S2: Visualization of the purification process. The first column is the clean stimuli, the second is the perturbed stimuli, and the rest are purified states.

