# OpenReview forum: "Dissecting Gradient Masking and Denoising in Diffusion Models for Adversarial Purification"
_ICLR.cc/2024/Conference — Submitted to ICLR 2024_

### Official Review · Reviewer_AzXU · 2023-10-29

**Soundness:** 2 fair
**Presentation:** 3 good
**Contribution:** 2 fair
**Rating:** 5
**Confidence:** 3

**Summary:**

Theoretically, we illustrate how a purification system with randomness can cause gradient masking, which can not be addressed by the standard expectation-over-time (EOT) method. Experimentally, we verify that gradient masking indeed happens under previous evaluations of diffusion models. This paper studies an interesting phenomenon in the adversarial purification based on diffusion models. However, I am still concerned about the experiments.

**Strengths:**

1. The idea of studying the connection between Diffusion-based adversarial purification and gradient masking is novel.
2. The theoretical result provide some new insights.

**Weaknesses:**

1. I am not fully convinced by the experiments that the diffusion-based adversarial purification causes the gradient masking. Here, the problem is whether there is measure of the gradient masking phenomenon.
2. The datasets are limited. Can the authors provide further experiments to verify their theoretical findings, i.e. performing experiments on other datasets, like Imagenet and MNIST?

**Questions:**

See the question above.

---

> ### Author Response · Authors · 2023-11-23
>
> We would like to thank the reviewer for the time spent reviewing our work. Also, we are pleased to hear that the reviewer recognizes the novelty and theoretical insights of the paper.
>
> Replies to the weaknesses:
> 1. Before answering the question, we want to first clarify that our theorem can be easily generalized to the case with simply connected adversarial space. This condition is practically appropriate, as Tramèr et al. 2017 showed that the adversarial space spans a continuous region with orthogonal bases. Both Reviewers 9RSr and AHc3 seem to have a correct understanding on this point. Assume we encounter randomness $\eta_0$ while calculating the attack. Then the attack is only optimal for the input with randomness $\eta_0$. During testing, however, we can encounter a different randomness $\eta_1$, then clearly the calculated gradient to generate the attack is not optimal (or useful) anymore under $\eta_1$. This is the main observation of how such a problem happens explained by our theorem.
> Besides theoretical arguments, we further add experimental results in the revised version to strengthen the claim. As shown in Fig. 2, noise has a major influence on the purified states in diffusion models. Starting from the same state, with different noise, the purified states will reach slight positively correlated directions (average correlation 0.3). However, even with different initial perturbations (average correlation 0) within the small ball, with the same noise, the purified states will be almost the same (average correlation > 0.99). Therefore, if not properly controlling the randomness, we are essentially applying the gradients with wrong states.
>
> 2. We added experiments on ImageNet in the modified version (Sec. 5.1 and 5.2). We used an unconditional guided diffusion model, with the ResNet-50 classifier. The main argument, randomness-induced gradient masking, was also identified in DiffPure on ImageNet (Table 1). Without randomness replay, our implementation of DiffPure achieves 69.0% clean accuracy, which is similar to the original reported 67.79% clean accuracy. After applying randomness replay, the clean accuracy stays to be a similar 68.5%, while the robust accuracy against BPDA drops to 34.0%, which is 33% lower than the non-randomness replay version. Note that after applying randomness replay, BPDA results provide a even better estimation than the full gradient-based AutoAttack, which is a clear sign of gradient masking in the original paper.
> The comparison of DiffPure and RevPure is a bit different on ImageNet. As shown in Fig 2c, unlike on CIFAR-10, the clean accuracy will drop by applying the reverse-only diffusion models. Therefore the best robust accuracy will be reached at a shorter timestep constraint by the drop of clean accuracy. However, the robust performances of DiffPure and RevPure are still similar. Therefore, our conclusion that RevPure can lead to a similar robustness improment also holds on ImageNet. A more comprehensive evaluation on ImageNet will be included in the final version.
>
> Finally, we want to emphasize that, given the popularity of EOT (3000+ citations) and DiffPure (130+ citations in a year), there exists an important issue in the appropriate evaluation of adversarial robustness involving randomness. Our paper provides a step toward addressing this important issue, both theoretically and empirically. There have been some interesting yet well-debated claims of robustness from randomness in both machine learning (Yoon et al. 2021, Byun et al. 2020) and neuroscience (Dapello, Joel, et al. 2020). There is an urgent need to shed light and clarify how randomness can (or can not) improve robustness. We hope our response and the revision will resolve your concerns and that you will re-assess the strengths and weaknesses of our paper.

---

### Official Review · Reviewer_gWaG · 2023-10-30

**Soundness:** 3 good
**Presentation:** 3 good
**Contribution:** 2 fair
**Rating:** 5
**Confidence:** 2

**Summary:**

This paper theoretically reveals the phenomenon that purification system with randomness provides false sense of robustness. A method named "randomness replay" is further proposed to better estimate the robustness. Experiments on Diffusion Models verify the proposals.

**Strengths:**

- The theoretical analysis is sound and easy to understand (e.g., illustration in Figure 1).
- The paper is easy to read.

**Weaknesses:**

- The structure of the paper is somewhat confusing. Diffusion models and gradient masking appear and mentioned in Abstract and Introduction, but most of the theoretical analysis are irrelevant with them.
- The experimental results shown in Tables seem weak. It will be better if more results are provided.

- Overall, this work is based on the DiffPure framework and present its one limitation. Moreover, attack on a system with randomness has been sudied. Therefore, I think the novelty is limited.

- Writing issue:
   - "Since the successful rate of applying the attack is the cdf of χ2(n), and n is the dimensionality of the data, which is typically very high for images (i.e., for CIFAR-10, n = 32 × 32 × 3 = 3072)."

**Questions:**

- It is mentioned "Gradient masking has been defined as “construct a model that does not have useful gradients”. Could you please explain how do you illustrate that a purification system with randomness can cause "a model does not have useful gradients"?
- It seems that the theoretical analysis are general and can be used for any preprocess-based defense methods. Could you please discuss more about the difference between this analysis and related work?
   - For example, Carlini etal. 2019 proposes "Verify that attacks succeed if randomness is fixed.". What is the difference between randomness replay and this method?

- Could you please provide more explanation if the forward and reverse process in Diffusion Models? Why DiffPure proposed to utilize both the two processes while only use the reverse process is also workable?

- Could you please explain that how does the observation that "The reverse process indeed gradually removes the projection on the original adversarial direction..." further support that "forward process may not be useful for robustness"? I think it will be better if you can also illustrate the forward process in Figure 4a.

---

> ### Author Response · Authors · 2023-11-23
>
> Thank you for your time in reviewing our work and for the critiques you raised. Below we address the reviewer's concerns. In particular, we would like to clarify a few points which seemed to have led to misunderstandings.
> Replies to the weaknesses:
> 1. The theoretical analysis in Sec. 4 is related and serves at the core part of our claim of gradient masking, without which we can not make the predictions at the end of Sec. 4 and further verify such predictions in Sec. 5. Both reviewer 9RSr and AHc3 seem to have a correct understanding on this point. Please hold a discussion with those reviewers if possible. More details in the reply to Question #1.
> 2. This is a valid concern. We added experiments on ImageNet in the modified version (Sec. 5.1 and 5.2). The main argument, randomness-induced gradient masking, was also identified in DiffPure on ImageNet (Table 1). After applying randomness replay, the clean accuracy remains similar, while the BPDA accuracy drops to 34.0%. With randomness replay, BPDA provides a better estimation than AutoAttack, which is a clear sign of gradient masking in the original paper.
> The comparison of DiffPure and RevPure is a bit different on ImageNet. Unlike CIFAR-10, the clean accuracy will drop by applying RevPure (Fig 2c, ). Therefore the best robustness will be reached with a shorter timestep constraint by the clean accuracy. However, the robust performances of DiffPure and RevPure are still similar. Our conclusion that RevPure can lead to a similar robustness improvement also holds on ImageNet.
> 3. The reviewer questions the novelty and significance of the work. We want to justify that our work in fact has substantial novelty and significance. Proposing new models or setting SOTA performance will be easily identified as novel, but we believe that analysis for correct understanding is also crucial. Given the popularity of EOT (3000+ citations) and DiffPure (130+ citations in a year), there exists an important issue in the evaluation of adversarial robustness involving randomness. Our paper provides a step toward addressing this important issue, both theoretically and empirically. There have been some interesting yet well-debated claims of robustness from randomness in both machine learning (Yoon et al. 2021, Byun et al. 2020) and neuroscience (Dapello, Joel, et al. 2020). There is an urgent need to shed light and clarify how randomness can (or can not) improve robustness.
>
> Replies to the questions:
> 1. We first clarify that our theorem can be easily generalized to simply connected adversarial space. This condition is practically appropriate, as Tramèr et al. 2017 showed that the adversarial space spans a continuous region with orthogonal bases.
> Assume we encounter randomness $\eta_0$ while calculating the attack. Then the attack is only optimal for the input with randomness $\eta_0$. During testing, however, we can encounter different randomness $\eta_1$, then clearly the calculated gradient to generate the attack is not optimal (or useful) anymore under $\eta_1$. Further experimental evidence is added in the modified version (Fig. 1).
> 2. Yes, there are no real differences between randomness replay and “fix randomness”. However, our analysis makes the argument more formal rather than just heuristicly proposing a method. Note that Carlini et al. were also in the author list of the EOT paper, which was widely used in adversarial purification and believed to solve the problem of randomness. We showed that EOT will not be sufficient, and randomness replay/fix randomness has to be the correct solution. Our results also explain why randomness matters—it is an inherent property of high-dimensional data due to the shape of the c.d.f of $\chi^2(n)$. When $n$ is large, both EOT and non-EOT are guaranteed to fail without randomness replay. In all, we believe our theoretical analyses provide a much deeper understanding of evaluating robustness involving randomness than just proposing a heuristic solution without knowing the causes and conditions.
> 3. This is another good question not directly answered by the DiffPure and DensePure papers. On CIFAR-10, the forward process is not necessary (Table 2 and Fig. 3). Especially through Fig. 3c, we can see that the forward process is not helpful for removing adversarial directions (the first flat stage)—adversarial directions are quickly removed only in the second reverse processing stage. For ImageNet, the forward process can be useful. Unlike on CIFAR-10, the clean accuracy drops by applying the RevPurfe, thus it is essential for the system to first inject a proper amount of noise for the reverse process to behave normally. However, the robustness performance of DiffPure and RevPure are still similar on ImageNet.
> 4. Again, this question can be answered by Fig. 3c—the forward process is not helpful for removing adversarial directions (the first flat stage)—adversarial directions are quickly removed only in the second reverse processing stage.

---

### Official Review · Reviewer_AHc3 · 2023-10-30

**Soundness:** 4 excellent
**Presentation:** 3 good
**Contribution:** 2 fair
**Rating:** 5
**Confidence:** 4

**Summary:**

Besides generating stunning images, diffusion models have been shown to be very suitable defense for adversarial robustness, which later became an area of research called "Adversarial Purification". This paper first theoretically distinguishes the role of individual components: randomness (gradient masking) of synthesis process and learned generative prior. Then they experimentally (on CIFAR-10) show that previous adversarial purification methods rely on gradient masking. Finally they propose new method RevPure that better controls the effect of randomness, and thus provides better robustness. Interestingly, authors also found that denoising process in diffusion models is orthogonal to original perturbation.

**Strengths:**

1) Solid theoretical analysis of the key mechanisms in diffusion-based adversarial purification methods. Illustrations and theorems are clear and interesting.
2) Easy-to-follow writing with good introduction, motivation, related work, and theoretical methodology.
3) Interesting observation that sequential denoising processes in diffusion models transform an adversarial example in the direction orthogonal to the original adversarial perturbation.
4) Interesting finding that forward process of diffusion models is unnecessary for adversarial purification, although it was observed previously in DensePure.

**Weaknesses:**

1) The closest works (DiffPure, DensePure) validated their experiments not only on CIFAR-10, but on large-scale datasets such as ImageNet. To make the method closer to real world, ImageNet experiments are necessary.
2) Only one architecture (WideResNet-28) is studied, while other works studied diverse set of architectures such as ViTs, ResNets, etc.
3) Effect of generation quality of the model (FID) on the robustness is only studied using two models (EMA DDPM and Non-EMA DDPM). You can't conclude anything from two points. More diverse set of models (maybe checkpoints from different epochs) should be studied.

Minor weaknesses:
1) $\chi^2$ should be properly introduced.

**Questions:**

1) "Consider we run an adversarial attack on a dataset and get an accuracy of 80%. Assume the attack is perfect, thus we can always find an adversarial example if there exists one in the region, which is the ultimate goal for adversarial attack research. Then the accuracy means for 80% of the data, we manage to find at least one adversarial example within the region. Thus, the empirical robustness with a perfect attack is a good approximation of the absolute robustness"
Can you elaborate more on this example? What is accuracy here (accuracy of the model on perturbed data or attack success rate)? How based on first 3 sentences we can claim that empirical robustness is a good approximation?
2) How does your method affect the inference time of the classifier?

---

> ### Author Response · Authors · 2023-11-23
>
> We thank the reviewers for the time and effort for reviewing our paper. We were glad to hear that the reviewer recognized the strength in theoretical analyses and the discovery of adversarial direction removal of diffusion models. Here are our replies to the raised critiques:
>
> 1. We added experiments on ImageNet in the modified version (Sec. 5.1 and 5.2). We used an unconditional guided diffusion model, with the ResNet-50 classifier. The main argument, randomness-induced gradient masking, was also identified in DiffPure on ImageNet (Table 1). Without randomness replay, our implementation of DiffPure achieves 69.0% clean accuracy, which is similar to the original reported 67.79% clean accuracy. After applying randomness replay, the clean accuracy stays a similar 68.5%, while the robust accuracy against BPDA drops to 34.0%, which is 33% lower than the non-randomness replay version. Note that after applying randomness replay, BPDA results provide an even better estimation than the full gradient-based AutoAttack, which is a clear sign of gradient masking in the original paper.
> The comparison of DiffPure and RevPure is a bit different on ImageNet. As shown in Fig 2c, unlike CIFAR-10, the clean accuracy will drop by applying the reverse-only diffusion models. Therefore the best robust accuracy will be reached at a shorter timestep constraint by the drop of clean accuracy. However, the robust performances of DiffPure and RevPure are still similar. Therefore, our conclusion that RevPure can lead to a similar robustness improvement also holds on ImageNet. A more comprehensive evaluation on ImageNet will be included in the final version.
> 2. In the modified version, ResNet-50 was added for ImageNet. Other classifier architectures WideResNet-50-2, DeiT-S could also be included in the final version (it’s just one line of code based on the RobustBenchmark). However, we believe diffusion models are essential to study in adversarial purification with diffusion models. We studied DDPM, unconditional Guided diffusion, as well as DDIM, which was not covered by DiffPure. We intentionally avoid any continuous-time diffusion models due to the potential extra gradient masking caused by the numerical solver (Huang et al. 2022).
> 3. It is a good suggestion to check multiple checkpoints with different FID scores, thus eventually changing Table 3 into a Figure. However, we want to serve our argument in Sec. 5.4 only as a counterexample—for anyone attempting to claim “diffusion models with a better image generation ability can achieve a better adversarial purification” in the future, they have to first explain our experiment in Table 3.
> 4. $\chi^2$: we add a “chi-square distribution” in theorem 1 in the modified version.
>
> Replies to the questions:
> 1. Sorry for the confusion about “accuracy” and “attack success rate”. The first “accuracy of 80%” will be corrected to “attack success rate of 80%” in the modified version to eliminate the ambiguity. Thanks for your responsible reviewing to point it out.
> For more elaboration, the absolute robustness for each data point is defined as a 0-1 variable (Definition 1). This paragraph of argument mimics the standard procedure of how we run the empirical adversarial attack (for instance, the PGD attack on CIFAR-10). The key is for each datum, we only apply the attack once, and either find an adversarial example or not. If the attack is perfect, an adversarial example should be found as long as it exists (no matter how small the adversarial region is). So, an 80% attack success rate for an ideal attack corresponds to a case where, for 80% of data, the absolute robustness is 1, and 20% is 0. Thus, it is a good approximation of the absolute robustness (taking the expectation over the dataset).
> 2. The main computational overhead comes from the reverse process of diffusion models, so there is no major difference between DiffPure and RevPure (which is indeed slightly faster). For the purification, since we are diffusing and denoising t* step, the inference time is approximately t*/T for generating an image with the diffusion model. For the classifier, we did not notify any influences on the inference time.
>
> Finally, we want to emphasize that, given the popularity of EOT (3000+ citations) and DiffPure (130+ citations in a year), there exists an important issue in the correct robustness evaluation involving randomness, which our paper provides both theoretical analyses and empirical solutions. Indeed, people were making false claims of robustness from randomness in both machine learning (Yoon et al. 2021, Byun et al. 2020) and neuroscience (Dapello, Joel, et al. 2020). We do think it is an important and urgent question to be corrected, otherwise those false claims of robustness will continually mislead the field. Hope the rebuttal and modifications will resolve your concerns and initiate a re-assessment of the contribution of the paper. Thanks!

---

### Official Review · Reviewer_9RSr · 2023-10-31

**Soundness:** 3 good
**Presentation:** 3 good
**Contribution:** 3 good
**Rating:** 5
**Confidence:** 3

**Summary:**

In this work, the authors revealed that diffusion models gain robustness from randomness, which causes gradient masking. They showed that randomness challenges the traditional expectation-over-time (EOT) method, leading to an overestimate of robustness. Addressing this, they proposed "randomness replay" for a more accurate robustness measure. They also found that robustness in diffusion models doesn't require a forward process; a reverse-only process is sufficient. Importantly, they demonstrated that diffusion models improve robustness by altering perturbed samples in a direction orthogonal to adversarial perturbations, thus weakening the strong attacks by removing the adversarial projections.

**Strengths:**

(1) The experimental results support the claim that randomness in the diffusion model can cause gradient masking, which cannot be addressed EOT method. The simple example in section 4.2 provides good intuition behind the claim.

(2) The exploitation of the purification power of diffusion by removing the adversarial projection is novel and insightful for future work.

**Weaknesses:**

(1) The study, while insightful, falls short of providing fundamental theoretical analysis. The example in section 4.2 is overly specific, leaving a gap in the broader understanding. A more general, perhaps probabilistic, theoretical exploration of how randomness leads to gradient masking and how randomness replay enhances robustness would significantly strengthen the work. Additionally, a theoretical validation of the diffusion model's ability to remove adversarial projections would be a valuable addition.

(2) The presentation of the theorems requires clearer explanations for each notation. For instance, in Theorem 1, the precise meaning of each symbol is unclear without delving into the proof. Providing meaningful explanations alongside the statement of each theorem would enhance comprehension.

**Questions:**

Can the authors explain why the diffusion model can remove adversarial projection? I believe this would be insightful for future work.

---

> ### Author Response · Authors · 2023-11-23
>
> We thank the reviewer for the constructive feedback and the effort for reviewing our submission. The reviewer acknowledged several strengths as well as the main contributions, namely identifying gradient masking and denoising in diffusion models for adversarial purification. The reviewer also pointed out several weaknesses of the initial submission which we have addressed in the revision. Below we provide detailed responses to the critiques raised by the reviewer:
>
> 1. Fundamental theoretical analysis. Our original submission considers a specific hyperball case, but indeed this example has a more general implication. As shown in theorem 1 of the modified submission, the theorem can be generalized to any classifier with a simply connected adversarial space. Note that the assumption of simply connected adversarial space is practically appropriate, as Tramèr et al. 2017 showed that the adversarial space spans a continuous region with orthogonal bases. The key observation of the theorem is that the problem caused by randomness in robustness evaluation is an inherent problem of high dimensional space, which can not be solved by either a stronger attack or EOT.
> We also state Corollary 2.1 more explicitly that randomness replay can overcome this problem. After randomness replay, the perfect attack will correctly estimate the absolute robustness (and EOT will have no benefits in this scenario), comparing with the non-randomness replay case, the success rate is approaching 0.
> Theoretical validation of adversarial projections removal in diffusion models. We provide more analysis in the new version, please refer to the reply of questions for more details.
>
> 2. Presentation of theorems. We apologize for any confusion in notations to potential readers. We indeed paid considerable attention to the clarity of the presentation in the original version: we tried our best to make all notations consistent throughout the entire paper, with definitions at the first time the notation appeared. The general structure is: we make the general setup in Sec. 4.1, then the setup for theorem 1 in Sec. 4.2, following the setup for theorem 2 in Sec. 4.3. In the modified version, we add instructions such as “Following the setup in Sec. 4.1” in each corresponding sections to increase the readability.  An explanation of “chi-square distribution” for $\chi^2(n)$ is also added in theorem 1 following the suggestion of reviewer AHc3.
>
> Replies to the questions:
> 1. Why can diffusion models remove adversarial projection? We added further analysis in the modified version to partially address this question (Fig. 4). Starting from a batch of random perturbations within a small ball around a clean stimulus, the reconstructed states after diffusion models are positively correlated with a much lower dimension structure. This shows that the learned image priors by diffusion models are biased toward a set of specific directions. Conceivably, these directions are unlikely to coincide with the adversarial directions. Otherwise, if the biased directions of diffusion models were indeed the adversarial directions,  the DiffPure method should fail even for clean images.
>
> In addition, we also added experiments on ImageNet corresponding to other reviewers' suggestions. Lastly, we want to emphasize that, given the popularity of EOT (3000+ citations) and DiffPure (130+ citations in a year), there exists an important issue in the correct robustness evaluation involving randomness, which our paper provides both theoretical analyses and empirical solutions. Indeed, people were making false claims of robustness from randomness in both machine learning (Yoon et al. 2021, Byun et al. 2020) and neuroscience (Dapello, Joel, et al. 2020). We do think it is an important and urgent question to be corrected, otherwise those false claims of robustness will continually mislead the field. Hope the rebuttal and modifications will resolve your concerns and initiate a re-assessment of the strengths and weaknesses of the paper.

---

### Meta-Review · Area_Chair_h9ww · 2023-12-05

**Metareview:**

This paper studies adversarial purification and its connection with gradient masking. The reviewers consensually considered that the experimental side of this paper was under the standard of the ICLR conference. In particular, it is not clear how the proposed insights and methods will scale to larger datasets. Even after the revision (where Imagenet experiments were added) the paper fall short in terms of experimental evidence. More complete experiments on more models and larger datasets would strengthen the paper.

**Justification For Why Not Higher Score:**

The reviewers consensually mentioned that the experiments were not strong enough.

**Justification For Why Not Lower Score:**

N/A

---

### Decision · Program_Chairs · 2024-01-16

Reject